# Fertility-Sparing Management May Be Considered in Young Women with Uterine Sarcoma

**DOI:** 10.3390/jcm11164761

**Published:** 2022-08-15

**Authors:** Szymon Piątek, Iwona Szymusik, Anna Dańska-Bidzińska, Mariusz Ołtarzewski, Gabriela Trojan, Mariusz Bidziński

**Affiliations:** 1Department of Gynecologic Oncology, the Maria Sklodowska-Curie National Research Institute of Oncology, 5 Roentgen Street, 02-781 Warsaw, Poland; 21st Department of Obstetrics and Gynecology, Medical University of Warsaw, 02-015 Warsaw, Poland; 32nd Department of Obstetrics and Gynecology, Medical University of Warsaw, 02-091 Warsaw, Poland; 4Institute of Mother and Child, Kasprzaka 17A Street, 01-211 Warsaw, Poland; 5Students’ Scientific Group, Kazimierz Pulaski University of Technology and Humanities in Radom, 26-600 Radom, Poland

**Keywords:** uterine sarcoma, fertility sparing, obstetric outcome, pregnancy, relapse

## Abstract

Uterine sarcomas occur very rarely in young women. Hysterectomy, which is a standard treatment, may not be acceptable for those patients, especially nulliparous women. Fertility-sparing management may be an alternative. The aim of the study was to assess fertility-sparing management in patients with uterine sarcoma. Eleven patients were eligible for the study. Histopathologic types of the tumor included: adenosarcoma (n = 3), low-grade endometrial stromal sarcoma (*n* = 3), low-grade myofibroblastic sarcoma (*n* = 1), leiomyosarcoma (*n* = 1), leiomyosarcoma myxoides (*n* = 1), rhabdomyosarcoma (*n* = 1), high grade endometrial stromal sarcoma (*n* = 1). The mean age of the patients at the time of diagnosis was 27.4 years (range: 17–35) and the average follow-up 61 months (range: 12–158). Six patients received adjuvant treatment: megestrol (*n* = 5) and chemotherapy (*n* = 1). Recurrence was diagnosed in five cases. Median time to recurrence was 35 months (range: 8–90). Three patients conceived spontaneously following treatment and gave at least one live birth. In total, five full-term pregnancies were recorded and five healthy children were born. Fertility-sparing management may be considered in some patients with uterine sarcoma; however, it may not be appropriate in high-grade endometrial stromal sarcoma. Patients with adenosarcoma may have a low chance of childbearing.

## 1. Introduction

Hysterectomy is the treatment of choice in the early stages of uterine sarcoma, while systemic chemotherapy is used in advanced cases. Uterine sarcoma occurs mainly in women over 40 years of age [1]. In those patients, fertility preservation is of less importance, while survival remains the main goal of treatment. In young women maintaining fertility may be crucial in making a therapeutic decision. Nowadays, fertility-sparing management has become one of the treatment options for young women with some malignancies. Among gynecological neoplasms, endometrial and cervical cancer are wellstudied and rules of management are included in the European Society of Gynaecological Oncology (ESGO) and National Comprehensive Cancer Network (NCCN) recommendations [2,3,4]. Much less is known about the fertility-sparing treatment of uterine sarcomas.

The diagnosis of uterine sarcoma is usually unexpected, because there are no specific radiological findings, no specific serum markers and no specific symptoms. In young patients, lesions may be incidentally found in the uterine cavity or within the myometrium, sometimes during infertility check-up. The suspicion of endometrial polyp/leiomyoma is an indication of a hysteroscopic resection or surgical excision but pathologic diagnosis of uterine sarcoma is devastating and usually ruins reproductive plans. Since the paper of Lissoni et al. [5] regarding fertility preservation in eight patients with leiomyosarcoma, only few articles have been published, mostly case reports [6,7,8]. The aim of this study was to assess the safety of fertility-sparing management and evaluate the number of successful pregnancies in patients with uterine sarcoma.

## 2. Materials and Methods

This retrospective study was conducted among patients treated between 1 January 2000 and 31 December 2021 in the Department of Gynecologic Oncology, Maria Sklodowska-Curie National Research Institute of Oncology (MSCNRIO). An electronic database (MSD) was searched using the term “uterine malignancy” and/or ICD-10 code “C54” and age “≤35 years ”. Medical records were reviewed by SP to find patients with uterine sarcoma, who underwent conservative treatment. Eligibility criteria for fertility-sparing management included: patient’s strong desire to preserve fertility, stage FIGO I/absence of extrauterine lesions on imaging, lack of residual disease in the uterus after tumor excision and age ≤ 35 years. The study flowchart is presented in Figure 1.

All patients underwent tumor excision in regional hospitals due to the suspicion of uterine leiomyoma: six patients had an abdominal lesion excision from the corpus of the uteri, three patients had a hysteroscopic tumor resection from the uterine cavity, two patients had a transvaginal tumor excision from the cervix of the uteri. Histopathological reports revealed uterine sarcoma or were inconclusive and patients were referred to our Institute. In all cases, histopathological re-evaluation was performed by expert pathologists in soft tissue/uterine sarcoma. After confirmation of the specific pathologic type of tumor, patients underwent imaging (pelvic magnetic resonance (MR), chest and abdominal computed tomography (CT)) to exclude metastasis, and a multidisciplinary consultation took place (obstetrician and gynecologist, gynecologic oncologist/surgical oncologist, clinical oncologist). Based on the postoperative MR and CT scans, patients were considered free of the residual disease and stage I was confirmed in all patients. All patients were fully informed about potential risks of radical and conservative management. All patients signed written consents for fertility-sparing management. Follow-up included gynecological examination with transvaginal ultrasonography every 3 months for up to 5 years after treatment. Additionally, patients with adenosarcoma underwent a hysteroscopy and endometrial curettage 6–12 months after the tumor resection. After 5 years and no signs of recurrence, patients were referred to a regional obstetrician and gynecologist.

Numbers and dates of deliveries after treatment were extracted from medical records and an electronic database at the Institute of Mother and Child, which includes information about all newborns in Poland using the 11-digit personal identification number (PESEL) assigned to all Polish citizens. Survival was verified in the PESEL electronic database, which contains the personal data of all Polish citizens.

The fertility-sparing surgery (FSS) was defined as sparing the uterus and ovaries. Pregnancies which ended in miscarriages were not taken into account.

## 3. Results

Eleven patients met the inclusion criteria. The mean age at the diagnosis was 27.4 years (range: 17–35 years). Ten (90.9%) patients were nulliparous. The most common histologic types of tumors were low-grade endometrial stromal sarcoma (LG ESS, 27.25%, *n* = 3) and adenosarcoma (ADS, 27.25%, *n* = 3), followed by high-grade endometrial stromal sarcoma (HG ESS 9.1%, *n* = 1), low = grade myofibroblastic sarcoma (9.1%, *n* = 1), leiomyosarcoma (9.1%, *n* = 1), leiomyosarcoma myxoides (9.1%, *n* = 1) and rhabdomyosarcoma (9.1%, *n* = 1). The patients’ characteristics are presented in Table 1.

All patients with LG ESS and two of the three patients with ADS received systemic gestagen therapy—megestrol acetate in daily single dose of 160 mg orally was administered for at least 12 months. Hormonal therapy was well tolerated, and no serious (grade 3/4) complications occurred. Patients with rhabdomyosarcoma were treated with chemotherapy—ifosfamide, vincristine and actinomycin D. Patients with leiomyosarcoma and one patient with adenosarcoma in the polyp did not receive adjuvant treatment.

The mean follow-up was 61 months (range: 12–158). Recurrence was found in five patients (45.45%), but none of the patients died during the follow-up period. Two patients decided to conduct follow-up in other units (patient no.1 after delivery and patient no. 6 due to personal difficulties in attending appointments). Mean time to recurrence was 38.8 months (range: 8–90). Four patients had a uterine recurrence and underwent a hysterectomy with bilateral salpingo-oophorectomy; one patient (with HG ESS) was diagnosed with ileus due to peritoneal carcinomatosis—she had an intestinal stoma and was qualified for palliative chemotherapy.

Three (27.25%) patients gave birth to a total of five children. One patient with LG ESS gave birth once 24 months after diagnosis, two patients gave birth twice—the patient with RMS delivered 41 and 71 months after diagnosis, while the patient with LG MS 12 and 34 months, respectively. All five pregnancies were delivered full term and no abnormalities in newborns were recorded.

## 4. Discussion

Uterine sarcomas are generally considered as poor prognosis malignancies. However, they comprise a diverse group of neoplasms with different courses and prognoses. Staging is the most important prognostic factor [9,10]. Histologic type [11] and mitotic index > 10(15) mitoses/10 HPF [12,13] are other well-established prognosis factors. Their rarity and histopathological diversity have contributed to the lack of consensus regarding their management [1].

Oncologic safety remains the major unknown in fertility-sparing treatment of uterine sarcoma. In our study, relapse was diagnosed in five patients, but none with adenosarcoma. Yuan et al., found one relapse among nine patients (11.11%) with adenosarcoma (four in the cervix, five in the corpus) [14]. Lee et al., reported two relapses in seven patients (28.57%) with adenosarcoma, but one of them had sarcomatous overgrowth at the time of diagnosis [15]. In another case report, the recurrence was diagnosed 8 years after the diagnosis [16]. All recurrences of ADS were localized in the uterus except for one patient with peritoneal seeding, who had a sarcomatous overgrowth at the time of diagnosis [15]. All patients with ADS were alive at the end of the studies period.

Our study reported one patient with HG ESS and fertility-sparing management. Although the disease was confined to a polyp, imaging was normal (pelvic MR, chest, and abdominal CT) and repeated hysteroscopy with dilatation and curettage revealed endometrium without any abnormalities, she was diagnosed with massive abdominal recurrence 8 months after the initial diagnosis. Due to ileus, she had an intestinal stoma and qualified for palliative chemotherapy. Recurrence was also diagnosed in two of three (66.66%) cases among patients with LG ESS. Similar recurrence rate (66.66%, four out of six patients) was reported by Tunc et al. [17]. Xie et al., also found a high rate of recurrence (58.82%, 10 of 17) in patients with LG ESS [18]. Different results in LG ESS were presented by Jin Y et al. [19] and Laurelli G et al. [20], who found 20% (one of five patients) and no recurrence, respectively. Results from case reports are also ambiguous. Ruixi Zhan et al. [21], Min Chul Choi [22] et al., Delaney et al. [8] reported patients who were free of disease, while Gu et al. [23], Payal S Jain [24], Tzu-Hsuan Chin et al. [25] presented cases with disease recurrence. Xie et al., suggested that tumor size (FIGO IA vs IB) may influence the recurrence rate [18]. Similarly to adenosarcoma, even after recurrence, all patients were alive apart from one patient, who died 10 years after FSS [26].

Leiomyosarcoma, the most common type of uterine sarcoma, is diagnosed less frequently in young women. In our study, one of two patients developed recurrence and was treated by hysterectomy. Lissoni et al., presented case series of eight patients with uterine leiomyosarcoma [5]. One patient (12.5%) had recurrence and died of a disseminated disease 26 months after the diagnosis. Tunc et al., found a recurrence rate as high as 71.4% (five out of seven cases) in patients with LMS [17]. Another fatal case was reported by Cormio et al. [27], who presented a 26-year-old patient with local recurrence 4 months after adjuvant chemotherapy and death 4 years after the diagnosis.

Adjuvant treatment after tumor excision remains unknown. Similar to our study, hormonal treatment with progestin (megestrol or medroxyprogesterone) was most commonly administered in patients with ADS and LG ESS [8,18,19,20,25,28,29,30]. However, the optimal dose and duration of progestin therapy was different among studies and varied from 80–100 mg/day [30,31] to 320 mg/day [19,32] of megestrol and 250 mg/day [21,33] to 600 mg/day [26,29] of medroxyprogesterone. Alternatively, analogues of gonadotropin [19], GnRH agonists [18], aromatase inhibitors [22] or a levonorgestrel-releasing intrauterine device [18] were used. Several patients with LG ESS received chemotherapy along with or instead of hormonal treatment [15,34]. Other authors reported successful treatment of patients with LG ESS without adjuvant treatment [23,35]. Patients in our study received hormonal treatment for 12 months, but in other reports the duration of the adjuvant progestin intake was shortened to 6 months [15,32]. Uterine leiomyosarcoma is a hormone-insensitive tumor with a questionable response to chemotherapy. Complete tumor removal could play a key role in preventing LMS recurrence, while the significance of any adjuvant treatment is not determined. Unlike LMS, all RMS cases should be treated with adjuvant chemotherapy [6,36,37,38].

Apart from survival, the number of successful pregnancies (i.e., pregnancies ending with deliveries) is the goal of fertility-sparing treatment. In our study, three (27.27%) patients conceived and a total of five children were born. The most recent systematic review regarding uterine preservation treatment in sarcomas was published by Dondi et al. (2021) [39]. The authors reported to have found 37 eligible papers, of which only four were case series, like our study. Among 210 women included in their review. 67 women (32%) conceived, of whom 17 patients (25%) required infertility treatment. The highest success rates with regard to conception (around 21%) were observed in cases with LG ESS, STUMP (smooth muscle tumor with uncertain malignant potential) and leiomyosarcoma. Patients with histological diagnosis of rhabdomyosarcoma (RMS) and adenosarcoma had a pregnancy rate of only 9% following FSS. Based on our results we also observed that patients with ADS were less likely to have a successful pregnancy. Three patients, who delivered, were diagnosed with different types of sarcoma—LG ESS, RMS and LG MS and none of the three with ADS conceived during the follow-up period. In another case-series including seven nulliparas with ADS, only a 33-year-old woman (14.29%) was able to conceive and deliver vaginally [15]. In studies involving other histological types than ADS, the successful pregnancy rate was higher. Xie et al., reported 17 patients with LG ESS, of whom eight attempted and five (29.41%) were able to conceive and deliver via cesarean section—four full-term and one preterm [18]. Laurelli et al., presented six patients with LG ESS aged 18–40 years, who underwent FSS in stage IA—three (50%) patients conceived and two (33%) of them delivered. Whereas Lissoni et al., reported three successful pregnancies among eight patients with leiomyosarcoma (37.5%) [5]. In our case series we reported a 22-year-old woman diagnosed with RMS of the cervix, who successfully attempted pregnancy twice and had two deliveries with no recurrence of the disease. Another rare clinical finding of low-grade myofibroblastic sarcoma in our case series referred to a 26-year-old woman, who also delivered twice following fertility-sparing treatment and remained disease-free for 90 months.

Reasons for lower pregnancy rates in patients with ADS and RMS remain unknown; however, low pregnancy rates in patients with RMS may be caused by multidrug chemotherapy and fertility impairment. Decreased fertility in patients with ADS may be due to repeated hysteroscopies and controlled D&C.

Since fertility-sparing options in uterine sarcomas are not the standard of care, the patients should always be thoroughly informed about the experimental treatment modality and the accompanying risk of disease recurrence. The available data suggest that in order for the procedure to be as safe as possible the tumor has to be completely resected with no sarcomatous overgrowth—therefore it can only reasonably be applied in stage I. Some authors underline that there also is no consensus regarding the follow-up of patients after fertility-sparing surgery and suggest monitoring them closely, including a hysteroscopy with an endometrial biopsy every 6 months and an MRI annually at least. It must also be underlined that patients receiving fertility-sparing surgeries should be encouraged to conceive as fast as possible due to the risk of recurrence. There seem to be no contraindications for infertility treatment, ovulation stimulation with letrozole (especially for those with estrogen receptor positive tumors) or in vitro fertilization procedures, as they may accelerate the chance of successful conception.

There are some limitations to this study. Firstly, the lack of infertility consultation in each case. In Poland, infertility treatment is unfortunately not fully reimbursed. The reproductive potential of the patients was therefore not assessed. Some patients may have needed assisted reproductive techniques due to alterations in theirs pelvic anatomy following surgery (such as post-operative adhesions, fallopian tube obstruction), but we had no such information. Secondly, data on miscarriages were not reported; therefore the total number of pregnancies may be underestimated. The study cohort of 11 patients may seem relatively small, but it is based on very rarely applied and unstandardized fertility-sparing management.

The available data on fertility-sparing surgeries in uterine sarcomas are still insufficient to make definitive judgements and recommendations. Close surveillance, long-term follow-up, and the possibility of post-delivery radical surgery, if necessary, should be mandatory. The patients should be managed by experienced multidisciplinary teams and their cases reported if possible. The more that is known, the better the counselling that can be provided with regard to fertility-sparing treatments. Wider knowledge may bring us closer to guideline development in those rare gynecological cases.

## Figures and Tables

**Figure 1 jcm-11-04761-f001:**
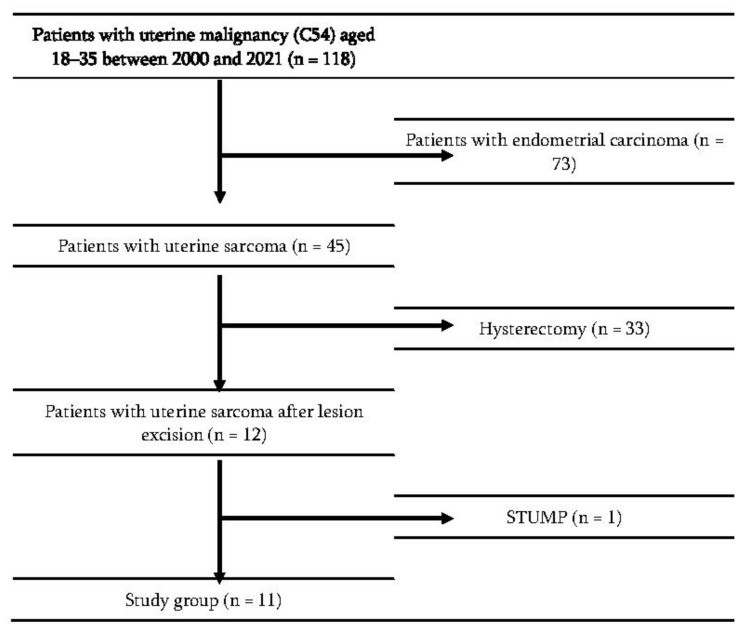
Study flowchart.

**Table 1 jcm-11-04761-t001:** Patients’ characteristics.

Patient	Age	Histology	Mitotic index (Mitoses Number/10HPF)	Nulliparous	Adjuvant Treatment	Follow-Up at the MSCNRIO (Months)	Recurrence/Time after Diagnosis (Months)	Delivery after Diagnosis
1	35	LG ESS	5	Yes	Hormone	12	No	1
2	17	ADS *	5	Yes	No	49	No	0
3	20	ADS **	9	Yes	Hormone	22	No	0
4	29	LG ESS **	low	Yes	Hormone	46	Yes/46	0
5	22	RMS *	-	Yes	Chemotherapy	59	No	2
6	33	ADS	3	Yes	Hormone	17	No	0
7	26	LG MS	low	Yes	No	158	Yes/90	2
8	21	LMS MX	7	Yes	No	143	Yes/15	0
9	34	LMS	10	Yes	No	67	No	0
10	34	LG ESS	6	Yes	Hormone	90	Yes/35	0
11	30	HG ESS **	high	No	No	12	Yes/8	0

ADS—adenosarcoma, HG ESS—high grade endometrial stromal sarcoma, LG ESS—low grade endometrial stromal sarcoma, LG MS—low grade myofibroblastic sarcoma, LMS—leiomyosarcoma, LMS MX—leiomyosarcoma myxoides, RMS-rhabdomyosarcoma.* tumor in the cervix of the uteri. ** polyp in the uterine cavity. low mitotic index means ≤10 mitoses/10HPV. high mitotic index means >10 mitoses/10HPV.

## Data Availability

The data presented in this study are available in the article.

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
