# Peer review of "Fertility-Sparing Management May Be Considered in Young Women with Uterine Sarcoma"

_jcm, 2022, doi:10.3390/jcm11164761_

Round 1

Reviewer 1 Report

The authors of “Fertility-sparing management may be considered in young women with uterine sarcoma” retrospectively evaluated 118 patients with uterine malignance at age =<35 years. From this group, they extrapolated 11 patients with uterine sarcoma stage I (FIGO) who underwent conservative treatment and desired to preserve fertility. All patients underwent tumor excision ( 6 -abdominal lesion excision; 3- hysteroscopic tumor resection from the uterine cavity; 2- transvaginal tumor excision from the cervix to the uteri. All cases had twice a histopathological evaluation, which confirmed uterine sarcoma after the additional exam by MRI and CT and exclude any metastasis. Multidisciplinary consultation and information about radical and conservative treatment risks have been performed. All patients signed written consents for fertility-sparing management. Ultrasound examination and gynecologist consultation have been done every 3 month up to 5 years after treatment. In cases of adenosarcoma  endometrial curettage was performed 6-12 month after treatment. After 5 years without any recurrence, patients have been referred to regional obstetrician&gynecologists. 

10 from 11 patients were nulliparous. The mean age 27.4 years. 3 cases low grade endometrial stromal sarcoma (160mg megestrole acetate daily for 12 month), 3 – adenosarcoma (for 2 cases 160mg megestrole acetate daily for 12 month/ 1-no adjuvant therapy ), 1-high grade endometrial stromal sarcoma, 1- low grade myofibroblastic sarcoma,1- leiomyosarcoma( no adjuvant treatment ),1- leiomyosarcoma myxoides,1-rhabdomyosarcoma (chemotherapy- ifosfamide, vincristine and actinomycin D) ( all well described in the figure1.) The mean follow-up was 61 months (range: 12-158). Recurrence was found in 5 patients (45.45%). 3 patients give birth to 5 children ( 1-once, 2 patients- twice), all full term without abnormalities.

I congratulate the authors for their work because fertility-sparing options in uterine sarcoma are not the standard of care and hysterectomy is usually a treatment of choice in the early stages of uterine sarcoma. Survival -remains a central goal of treatment in oncology, but according to this work,  staging and histological type are essential factors as a predictor of surveillance and fertility preserving opportunities.

I suggest this paper with minor revision 

1) Except well-shown information in figure 1 to describe in the RESULTS who and after how many years  these patients gave birth. 

2) To describe clear follow-up for each case because the range is too large (12-158 month) but also because of misunderstanding in the MATERIAL and METHODS have been written that cases followed-up 5 years after treatment with consequent referral to the regional OB&GYN. 

3) Some mispellings  of English . Please review the manuscript with a English native .

Author Response

I suggest this paper with minor revision 

1) Except well-shown information in figure 1 to describe in the RESULTS who and after how many years  these patients gave birth. 

2) To describe clear follow-up for each case because the range is too large (12-158 month) but also because of misunderstanding in the MATERIAL and METHODS have been written that cases followed-up 5 years after treatment with consequent referral to the regional OB&GYN. 

3) Some mispellings  of English . Please review the manuscript with a English native .

Thank you very much for your time and valuable comments. We tried to address all of them the best we could. We truly hope that the provided data is satisfactory.

1) We added some information in Results section regarding the deliveries (“who and after how many years delivered”) – they are attached between lines 121-124: 1 patient with LG ESS gave birth once 24 months after diagnosis. 2 patients gave birth twice – patient with RMS delivered 41 and 71 months after diagnosis, while patient with LG MS 12 and 34 months, respectively.

2) We added information with clear follow-up time for each case of sarcoma – additional column was added to the Table (follow-up at National Research Institute of Oncology in months) – it was marked in yellow. Material and methods section describes the plan for follow-up in cases of sarcoma suggested by the Institute. However, not all the patients attended the appointments at our center – some of them decided to choose centers closer to their homes. The number of follow-up months only corresponds to the actual appointments at our NRI. Nevertheless, all the patients were alive at the time of manuscript preparation, as we verified PESEL database for each case. We added new sentence to the manuscript (lines 109-111): Two patients decided to conduct follow-up in other hospitals (patient no. 1 after the delivery and patient no 6. due to personal difficulties in attending appointments).

3) English language corrections were also introduced and the paper was once again reviewed with regard to misspellings

Reviewer 2 Report

This retrospective study on fertility-sparing management in patients with uterine sarcoma describes the outcomes of a small cohort of 11 patients undergoing fertility-sparing therapy for different types of uterine sarcoma. The manuscript is well written and clearly structured. The authors report real-world data with a moderate follow-up. In the discussion section, the authors should also report the limitations of the present study. 

Author Response

This retrospective study on fertility-sparing management in patients with uterine sarcoma describes the outcomes of a small cohort of 11 patients undergoing fertility-sparing therapy for different types of uterine sarcoma. The manuscript is well written and clearly structured. The authors report real-world data with a moderate follow-up. In the discussion section, the authors should also report the limitations of the present study. 

Thank you very much for your time and valuable comments. According to your advice a short paragraph on study limitations was added to the discussion section of the manuscript and was marked in yellow (lines 242-249): There are some limitations of this study. Firstly, the lack of infertility consultation in each case. In Poland infertility treatment is unfortunately not fully reimbursed. The reproductive potential of the patients was therefore not assessed. Some patients might have needed assisted reproductive techniques due to alterations of pelvic anatomy following surgery (such as post-operative adhesions, fallopian tube obstruction), but we had no such information. Secondly, data on miscarriages were not reported, therefore the total number of pregnancies may be underestimated. The study cohort of 11 patients might seem relatively small, but it is based on very rarely applied and not standardized fertility-sparing management.

Reviewer 3 Report

The discussion is too long and confusing, I suggest restructuring it and focus on highlighting the most important aspects of the study.

Author Response

The discussion is too long and confusing, I suggest restructuring it and focus on highlighting the most important aspects of the study.

Thank you very much for your time and valuable comments. After reviewing the manuscript once again we decided not to shorten the discussion. We would like to address why: Uterine sarcomas are a diverse group of malignancies with different prognosis and treatment. In our material patients were diagnosed with different types of uterine sarcomas.

We have discussed the results for each type separately, as prognosis and treatment differs among each type – therefore it should not be confusing, but seems to be long. Moreover, most of the publications on fertility sparing surgeries in uterine sarcomas are based on case reports, where authors presented various observations. To avoid a one-sided presentation of data, we cited many publications. We believe that this way of confronting our results with other publications enables readers to have the broadest possible view on fertility sparing treatment of uterine sarcomas. We truly hope you will find our explanation satisfying.

Reviewer 4 Report

Please state that among the 3 recorded pregnancies, 2 were from the same patient

Author Response

Please state that among the 3 recorded pregnancies, 2 were from the same patient

Thank you very much for your time and valuable comments. We stated clearly that 5 pregnancies were recorded in 3 different patients (“3 (27.25%) patients gave birth to a total of 5 children. One patient gave birth once, 2 patients gave birth twice. All 5 pregnancies were delivered full term and no abnormalities in newborns were recorded”). We additionally included data regarding the time from diagnosis to the delivery in months in each case (lines 126-128).

This manuscript is a resubmission of an earlier submission. The following is a list of the peer review reports and author responses from that submission.